# Comparative Analysis of Serum N-Glycosylation in Endometriosis and Gynecologic Cancers

**DOI:** 10.3390/ijms26094105

**Published:** 2025-04-25

**Authors:** Róbert Pásztor, Béla Viskolcz, Csaba Oláh, Csaba Váradi

**Affiliations:** 1Institute of Chemistry, Faculty of Materials and Chemical Engineering, University of Miskolc, 3515 Miskolc, Hungary; csuko67@gmail.com (R.P.); bela.viskolcz@uni-miskolc.hu (B.V.); 2Department of Neurosurgery, Borsod-Abaúj-Zemplén County Center Hospital and University Teaching Hospital, 3526 Miskolc, Hungary; olahcs@gmail.com

**Keywords:** myoma uteri, endometriosis, cervical carcinoma, N-glycosylation, biomarker discovery

## Abstract

Gynecologic tumors are a leading cause of cancer-related mortality in women worldwide, with endometrial, ovarian, and cervical types being the most prevalent. Aberrant glycosylation, a key post-translational modification, plays a crucial role in tumor development, metastasis, and immune evasion. Specific glycosylation changes, such as altered sialylation and fucosylation, have been identified in gynecologic cancers and are associated with disease progression and prognosis. Understanding glycosylation alterations in gynecologic cancers holds promise for novel diagnostic and therapeutic approaches, ultimately enhancing patient outcomes. In this study, the serum N-glycome was analyzed in patients with myoma uteri, endometriosis, and cervical carcinoma by hydrophilic-interaction liquid chromatography (HILIC-UPLC) with fluorescence (FLR) and mass-spectrometric (MS) detection in order to identify their biomarker potential. Individual serum samples were deglycosylated by PNGase F digestion followed by procainamide labeling and solid-phase-extraction-based purification. All disease groups exhibited consistently higher levels of specific bi-antennary glycans (A2G2 and A2G2S1) compared to control patients. Additionally, significantly higher levels of agalactosylated and mono-sialylated glycans were found in cervical cancer, while a notable decrease in bisected N-glycans, alongside an increase in highly branched tetra-sialylated glycans, was found in endometriosis. Our study serves as proof-of-concept, demonstrating that discovering biomarkers within the serum N-glycome is a promising approach for identifying non-invasive indicators of gynecologic conditions.

## 1. Introduction

Gynecologic cancers are among the leading causes of cancer-related deaths in women worldwide [1]. Cervical cancer remains a significant health concern, with its progression closely linked to human papillomavirus (HPV) infection [2]. Despite advancements in screening and vaccination, cervical cancer continues to contribute substantially to gynecologic cancer morbidity and mortality [3]. Glycosylation involves the enzymatic addition of sugar moieties to proteins, influencing their structure and function [4]. In cancer, aberrant glycosylation patterns can promote malignant transformation, tumor development, invasiveness, and metastasis [5]. The early detection of gynecologic cancers is crucial for improving patient outcomes as current diagnostic tools often lack sensitivity and specificity, especially for early-stage disease [6]. Studies have shown distinct sialylation and fucosylation patterns in cervical cancer tissues compared to normal tissues, suggesting that these glycosylation changes play a role in the disease’s development and progression [7]. In gynecological diseases, particularly gynecologic cancers such as endometrial, ovarian, and cervical cancers, aberrant glycosylation patterns have been observed and are associated with disease progression and prognosis [5]. Understanding these glycosylation alterations is essential for improving diagnosis, assessing disease outcomes, and developing effective biomarkers [8]. The identification of reliable biomarkers is essential to enhance early diagnosis, monitor disease progression, and predict treatment responses [9]. The field of glycoproteomics, focusing on the comprehensive study of glycosylated proteins, has advanced significantly with the development of mass spectrometry techniques [10]. These advancements have facilitated the identification and characterization of glycosylation patterns, aiding in the discovery of novel biomarkers for gynecologic cancers [11]. High-resolution mass spectrometry allows for the detailed analysis of glycan structures and their attachment sites on proteins, providing insights into disease mechanisms and potential therapeutic targets [12]. Understanding these glycosylation changes is vital for developing effective diagnostic tools and identifying reliable biomarkers [13]. Advancements in glycoproteomics offer promising avenues for early detection, prognosis assessment, and personalized treatment strategies in gynecological oncology [14]. Altered glycosylation patterns, such as increased sialylation and reduced fucosylation, have been observed in cervical cancer and associated with tumor progression [15].

In this study, the serum N-glycome was analyzed in patients with myoma uteri, endometriosis, and cervical carcinoma by HILIC-UPLC (hydrophilic-interaction liquid chromatography) with FLR (fluorescence) and MS (mass-spectrometric) detection to determine the biomarker potential of protein glycosylation in these diseases. Individual serum samples were deglycosylated by PNGase F digestion followed by procainamide labeling and solid-phase-extraction-based purification. Significant differences were identified in each disease group compared to healthy controls.

## 2. Results and Discussion

### 2.1. Laboratory Data Analysis

Samples from 22 healthy controls, 22 myoma uteri patients, 22 endometriosis patients, and 22 cervical carcinoma patients were analyzed. Seventeen laboratory parameters were measured at baseline to characterize each group (Table 1). The endometriosis group was slightly younger on average (40.1 ± 4.3 years) than the other groups, consistent with literature indicating that endometriosis primarily affects women of reproductive age. Renal function markers (urea, creatinine, eGFR) showed no significant differences across groups (all *p* > 0.2), suggesting no major renal impairment, in line with expectations that renal function remains normal unless severe uterine enlargement causes obstruction. As expected, hemoglobin was lower in myoma uteri patients (reflecting possible chronic menstrual blood loss), while platelet counts were significantly higher in that group. Inflammatory markers differed: C-reactive protein (CRP) was elevated in endometriosis (median ~10 mg/L vs. ~3 mg/L in controls), confirming a chronic inflammatory state, and alkaline phosphatase (ALP) was higher in cervical carcinoma patients, supporting known patterns of malignancy-associated enzyme elevation. These baseline differences confirmed the distinct clinical contexts of each group and underscore the importance of adjusting for potential confounders in subsequent analyses.

### 2.2. Analysis of Serum N-Glycome

We performed a comprehensive serum N-glycome analysis using HILIC-UPLC with fluorescence detection, supported by MS for structural identification. In total, 14 N-glycan structures showed significantly different relative abundances among the four groups by the Kruskal–Wallis test (Table 2 for full statistics). Figure 1 illustrates a representative HILIC-FLR chromatogram of serum N-glycans, highlighting the most abundant structures (bi-, tri-, and tetra-antennary glycans with varying fucosylation and sialylation). The most pronounced disease-associated glycosylation changes were observed when comparing each patient group to healthy controls, as well as between certain disease groups. The collected serum samples were analyzed by HILIC-FLR-MS, where fluorescence data were used for quantitation and mass-spectrometric data were used for structure identification. Representative chromatographic separation is visualized in Figure 1, highlighting the most abundant glycan structures.

The generated data were used for statistical tests to find correlations and significant differences between the four patient groups. In the case of 14 structures, significantly different ratios were detected by the Kruskall–Wallis test, as shown in Table 2. The most significant differences in relative glycan distributions across the examined groups are visualized in Figure 2A–I. In the case of several glycan structures, systematic shifts were identified when comparing control vs. disease and disease vs. disease. For example, the level of FA2 (fucosylated bi-antennary with no galactose, analogous to G0F) (Figure 2A) in the disease groups was in the same range compared to the control, while in cervical carcinoma, it was found to be significantly higher than in myoma uteri and endometriosis, suggesting increased agalactosylation in malignancy. Conversely, the fully galactosylated bi-antennary glycan FA2G2 (G2F) tended to be lower in the cancer group (Table 2). The consistent rise of agalactosylated N-glycans in cervical carcinoma is noteworthy, as elevated G0F glycans have been reported in various cancers [16]. The relative ratio of FA2B (Figure 2B) was significantly lower in endometriosis compared to the control and cervical carcinoma groups, which aligns with recent findings that structures with bisecting N-acetyl-glucosamine are decreased in endometriosis [17]. In contrast, FA2(3)G1 (a mono-galactosylated, core-fucosylated bi-antennary) showed a significant difference only between myoma uteri and cervical carcinoma (*p* < 0.05), hinting that partial galactosylation changes might distinguish benign fibroids from malignant tumors (Figure 2C). The relative ratio of A2G2 (non-fucosylated, bi-antennary with two galactoses) was elevated in all three disease groups (Figure 2D) compared to controls (median relative abundance ~0.85–1.03 in diseases vs. ~0.70 in controls, *p* < 0.01 for each), representing a ~40% increase in disease states. The effect was most pronounced in cervical carcinoma (mean difference ~0.33, 95% CI of difference ≈0.20–0.45), highlighting a large effect size. This glycan (A2G2) lacks core fucosylation, suggesting a shift toward non-fucosylated glycoforms in disease. Indeed, decreased core fucosylation has been observed in cervical cancer tissues [18].

A2G2S1 (a bi-antennary, sialylated glycan) was higher in all disease groups (Figure 2E) compared to the control group (median in control ≈10.4 vs. ≥11.5 in diseases); this was in contrast to the results in Figure 2G,H, where significantly lower levels of sialylation were detected on bi- and tri- antennary structures. Interestingly, while A2G2S1 showed higher levels in the disease groups, FA2G2S1 was rather decreased, which was also significant in the group of myoma uteri (Figure 2F). Cervical carcinoma vs. control differences were most pronounced in glycans, e.g., A2G2 and A2G2S1 were much higher in the carcinoma group. Notably, A2G2 and A2G2S1 reached significance in all three disease groups versus the control, underscoring these as consistently altered features. Other glycan differences were more specific: the tetra-antennary glycans A4G4S4 (Figure 2I) were significantly elevated in endometriosis (means ~0.60 vs. 0.50 in control) but slightly decreased in myoma and cervical carcinoma, explaining the overall significance and suggesting opposite trends by disease. The Kruskall–Wallis results confirmed that glycan profile changes are a hallmark of all three diseases, with A2G2 and A2G2S1 emerging as the most significantly altered (these two features differed by ~40% or more between the control and diseases).

Overall, the glycosylation changes identified were substantial. Many of the glycan traits showed differences with large effect sizes (percentage changes of 20–50% between medians of patients and controls). For the most consistently altered features (A2G2 and A2G2S1), the non-overlapping interquartile ranges between the control and disease groups underscored their potential as robust biomarkers. In summary, all three gynecologic disease groups (including the benign myoma and endometriosis) demonstrated distinct serum N-glycan alterations relative to the healthy individuals. Cervical carcinoma, being malignant, showed the most extreme changes (e.g., pronounced increase in A2G2 and A2G2S1 and a cancer-specific elevation of agalactosylated FA2). Endometriosis and myoma uteri, while not malignant, still exhibited notable glycomic shifts (particularly in sialylation and bisected structures), suggesting that even benign gynecological conditions are accompanied by systemic glycosylation changes. These disease-associated glycan signatures provide a foundation for developing glycan-based biomarkers and deepen our understanding of the underlying pathophysiology.

To further interrogate group separation, we applied linear discriminant analysis (LDA), a supervised method, using the glycan variables to maximize discrimination between the four groups. The scatter plot in Figure 3A represents the projection of different disease groups onto two LDA components, which were the axes that best separated the groups. Each point corresponds to a sample, and its position was determined by the glycan composition. The disease groups appear in distinct regions of the plot, indicating that LDA successfully captured the variance in glycan profiles. The primary discriminant function (LDA Component 1) explained 74.18% of the variance and provided the strongest separation between groups. The secondary discriminant function (LDA Component 2) explained 21.45% of the variance, adding additional but less dominant separation. The endometriosis and myoma uteri groups had partial overlap, suggesting shared glycan characteristics or less distinct differences. Based on LDA loadings, the top glycans that influence each disease group the most were as follows:➢Cervical carcinoma: FA2G1S1, A3G3S3 (3), FA3G3S3 (3)➢Control: FA2G1S1, A3G3S3 (3), M5➢Endometriosis: A3G3S3 (1), A4G4S3, FA2BG2➢Myoma uteri: A3G3S3 (1), FA2BG2, FA2G1S1

These glycans had the highest contribution in defining the boundaries between the disease groups and are likely key biomarkers for classification. The overall importance of the separation of the examined groups is visualized in Appendix A, based on the sum of absolute LDA coefficients. Finally, we evaluated the potential of the altered glycans as biomarkers by performing receiver operating characteristic (ROC) curve analyses, as shown in Figure 3B. For each disease group vs. healthy control, we computed ROC curves for individual glycan features as well as for combinations. The area under the ROC curve (AUC) was used to quantify discrimination performance (AUC = 0.5 indicated no discrimination; AUC = 1.0 indicated perfect classification). Overall, several glycan features showed promising diagnostic performance. For distinguishing cervical carcinoma from controls, the top single glycans were A2G2 and A2G2S1, each with an AUC of ≈ 0.85–0.90. Specifically, A2G2 had an AUC of about 0.88 (95% CI ~0.77–0.98), reflecting its high levels exclusively in cancer patients (Figure 3B). A2G2S1 had an AUC of ~0.82, nearly as informative (Appendix A). Additionally, the agalactosylated FA2 (G0F) yielded an AUC of ~0.80 for cervical cancer vs. control—consistent with its elevation in cancer—and the decrease in FA2G2S1 (a fucosylated monosialylated glycan) in the myoma group meant that glycan was less useful for cancer detection but did contribute to distinguishing myoma (Appendix A). By contrast, for endometriosis, no single glycan reached such high AUC values, although the uniquely increased A4G4S4 in endometriosis also achieved an AUC of ~0.70. These values indicate modest discriminative power individually. The benign myoma uteri group was the hardest to distinguish by any single glycan; features like A2G2S1 and A2G2 (which did differ vs. controls) still only gave an AUC of ~0.70–0.75 for myoma vs. control, perhaps because those changes in myoma, though significant, were less pronounced than in the carcinoma. In general, multivariate panels of 3–5 glycans pushed AUC values into the 0.85–0.95 range for all three diseases. This suggests that while no single glycan is a perfect biomarker for the benign conditions, a composite glycan signature could be quite powerful.

**Figure 3 ijms-26-04105-f003:**
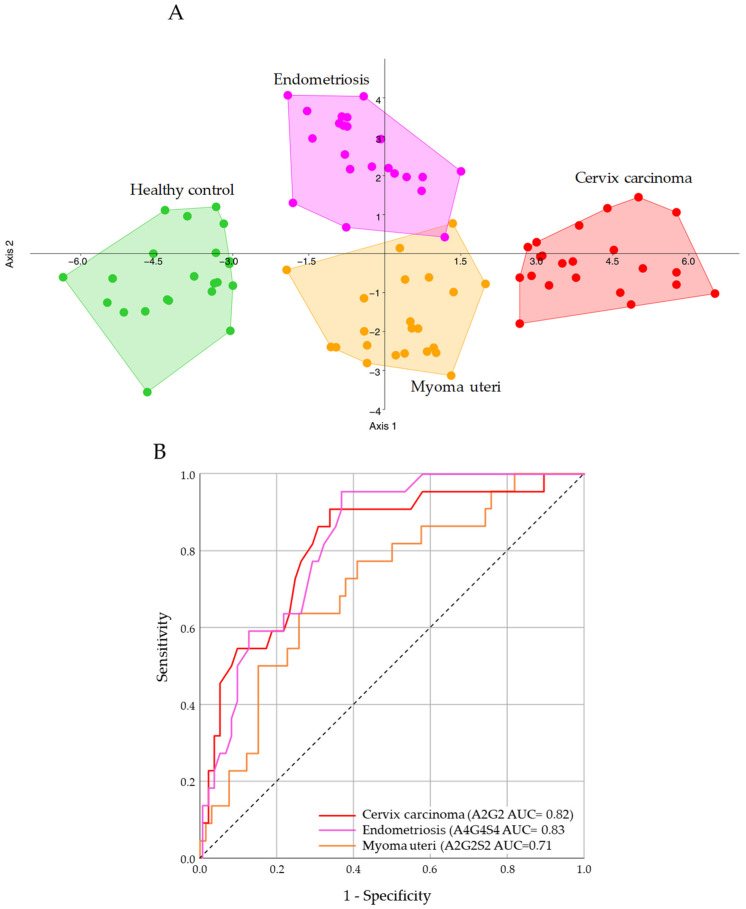
Linear discriminant analysis (**A**) and ROC curve analysis (**B**) of serum N-glycome from healthy control (green), myoma uteri (orange), endometriosis (pink) and cervical carcinoma (red) patients.

A strength of our approach is the use of a uniform analytical workflow (UPLC-FLR and MS identification) across all samples, which ensured that the differences observed were biologically driven rather than technical. Including multiple disease groups enabled internal comparisons that strengthen confidence in the disease-specific findings. However, this study has limitations: the sample size in each group (n = 22) was moderate, which may have limited the power to detect smaller differences and led to the overestimation of some effect sizes. The cross-sectional nature meant that we could not determine if the observed glycan changes were causes or consequences of the diseases. Another limitation is the lack of an independent validation cohort; our findings, though statistically significant, would need confirmation in larger, independent groups of patients to establish generalizability. Despite these limitations, the present study provides a strong proof-of-concept that serum N-glycan patterns differentiate between health and various gynecologic pathologies.

The demonstrated glycosylation differences carry several implications suggesting that systemic glycan alterations are intimately linked with gynecologic disease processes. This opens avenues to use glycans as biomarkers: for example, a blood test measuring a panel of glycans (such as A2G2, A2G2S1, FA2B, A4G4S4) could aid in early detection or screening.

## 3. Materials and Methods

### 3.1. Chemicals

Formic acid, ammonium hydroxide, acetic acid, acetonitrile, picoline borane, procainamide hydrochloride, and dimethyl sulfoxide were sourced from Sigma-Aldrich (St. Louis, MO, USA). PNGase F was purchased from New England Biolabs (Ipswich, MA, USA).

### 3.2. Patient Samples

Serum samples were collected from 88 patients categorized into four groups: 22 healthy controls, 22 with myoma uteri, 22 with endometriosis, and 22 with cervical carcinoma (stage 0: carcinoma in situ). The samples were obtained at Borsod Academic County Hospital (Miskolc, Hungary). All disease groups were confirmed through histological analysis, as each patient underwent surgical treatment. Patients were selected based on the absence of comorbidities, and healthy controls reported no gynecological complaints. The preparation of serum samples is a critical step in various laboratory analyses, particularly in clinical and research settings. Blood samples were collected using sterile vacutainer tubes without anticoagulants. After collection, the samples were allowed to clot at room temperature for approximately 30 min, facilitating the separation of serum from cellular components. Subsequently, samples were centrifuged at 2000 RPM for 10–15 min. The serum was carefully pipetted into new sterile tubes, ensuring that the clot remained undisturbed. For storage, samples were frozen at −80 °C, depending on the requirements of the intended analysis. Table 1 summarizes the baseline characteristics of the patient samples. This study received approval from the Regional Research Ethics Committee (ethical approval number: BORS-19/2023), and all patients provided written informed consent in accordance with the Declaration of Helsinki.

### 3.3. N-Glycan Release from Serum Proteins, Labeling, and Clean-Up

N-glycan release was conducted following the PNGase F deglycosylation protocol from New England Biolabs (Ipswich, MA, USA) using 9 µL of serum sample. The released carbohydrates were fluorescently labeled by adding 10 μL of a 0.3 M procainamide and 300 mM picoline borane solution in a 70%/30% dimethyl sulfoxide/acetic acid mixture, followed by incubation at 65 °C for 4 h. The labeled glycans were then purified using NH_2_-functionalized MonoSpin columns (GL Sciences Inc., Tokyo, Japan) according to the manufacturer’s instructions. The purified glycans were dissolved in a 25%/75% water/acetonitrile solution and analyzed using HILIC-UPLC-FLR-MS.

### 3.4. UPLC-FLR-MS Analysis

Fluorescently labeled N-glycans were analyzed using a Waters Acquity ultra-performance liquid chromatography (UPLC) system equipped with a fluorescence detector and Xevo-G2S qTOF mass spectrometer controlled by MassLynx 4.2 software (Waters, Milford, MA, USA). Separation was performed on a Waters BEH Glycan column (100 × 2.1 mm i.d., 1.7 μm particle size) with a linear gradient of 75–55% acetonitrile (Buffer B) at a flow rate of 0.4 mL/min over 22 min, using 50 mM ammonium formate (pH 4.4) as Buffer A. Each run used a 1 μL injection volume, with the sample manager set at 15 °C and the column maintained at 60 °C. Fluorescence detection was performed with excitation and emission wavelengths of *λ*ex = 308 nm and *λ*em = 359 nm, respectively. Mass spectrometry (MS) analysis was conducted in positive ionization mode, applying a 3 kV electrospray voltage to the capillary. The desolvation temperature was set to 120 °C, with a desolvation gas flow rate of 800 L/hr. Mass spectra were acquired over the 500–3000 *m*/*z* range.

### 3.5. Data Analysis

Chromatograms of the patient samples were integrated using Unifi chromatography software (Waters, Milford, MA, USA) based on fluorescence spectra, with MS confirmation. The mass-to-charge ratios of individual glycan structures were determined using GlycoWorkbench 2.0. Statistical analyses, including the Kruskal–Wallis test and Mann–Whitney pairwise comparisons, were performed using IBM SPSS Statistics 23. Linear discriminant analysis was conducted with Past 4.11 software. Figures were generated using GraphPad Prism 10.

## 4. Conclusions

In this study, we conducted a comparative analysis of serum N-glycosylation profiles in healthy women versus those with endometriosis, uterine fibroids (myoma uteri), and cervical cancer. The results revealed distinct glycan signatures for each condition, highlighting the profound ways in which systemic glycosylation is altered in gynecologic diseases. Key findings include consistently elevated levels of specific bi-antennary glycans (A2G2 and A2G2S1) in all disease groups compared to controls, pronounced increases in agalactosylated and mono-sialylated glycans in cervical cancer, a unique decrease in bisected N-glycans in endometriosis, and an increase in highly branched tetra-sialylated glycans in endometriosis. These alterations were statistically significant, with large effect sizes and non-overlapping confidence intervals for major differences, underscoring their biological importance. This opens the door to less invasive diagnostic tools: for example, a serum glycan test might help detect or monitor endometriosis, a disease that currently requires surgical diagnosis. In the realm of oncology, glycan biomarkers could improve the early detection of gynecologic cancers or help distinguish malignant from benign pelvic masses. Future research should focus on larger cohorts and prospective designs to confirm these biomarkers, as well as experimental studies to unravel the causal links between glycosylation changes and disease pathology. Ultimately, integrating glycomic biomarkers with existing diagnostic workflows could enhance early detection, personalize patient management, and lead to better outcomes in gynecologic diseases. The prospect of using a simple blood test to detect conditions like endometriosis or distinguish benign from malignant ovarian masses is an exciting translation of glycoscience to precision medicine in gynecology.

## Figures and Tables

**Figure 1 ijms-26-04105-f001:**
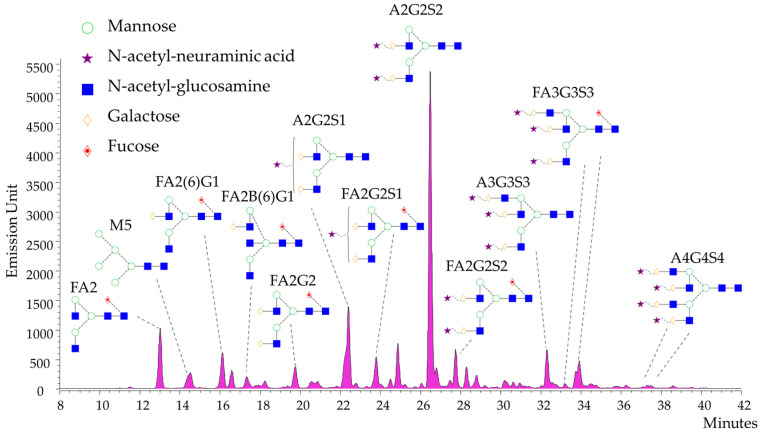
Representative fluorescence chromatogramm of serum N-glycans by HILIC-FLR (structural nomenclature: FA2: fucosylated bi-antennary, FA2G1: fucosylated and mono-galactosylated bi-antennary, FA2G2: fucosylated and bi-galactosylated bi-antennary, A2G2S1: mono-sialylated and bi-galactosylated bi-antennary, A2G2S2: bi-sialylated and bi-galactosylated bi-antennary, A3G3S3: tri-sialylated and tri-galactosylated bi-antennary, A4G4S4: tetra-sialylated and tetra-galactosylated tetra-antennary).

**Figure 2 ijms-26-04105-f002:**
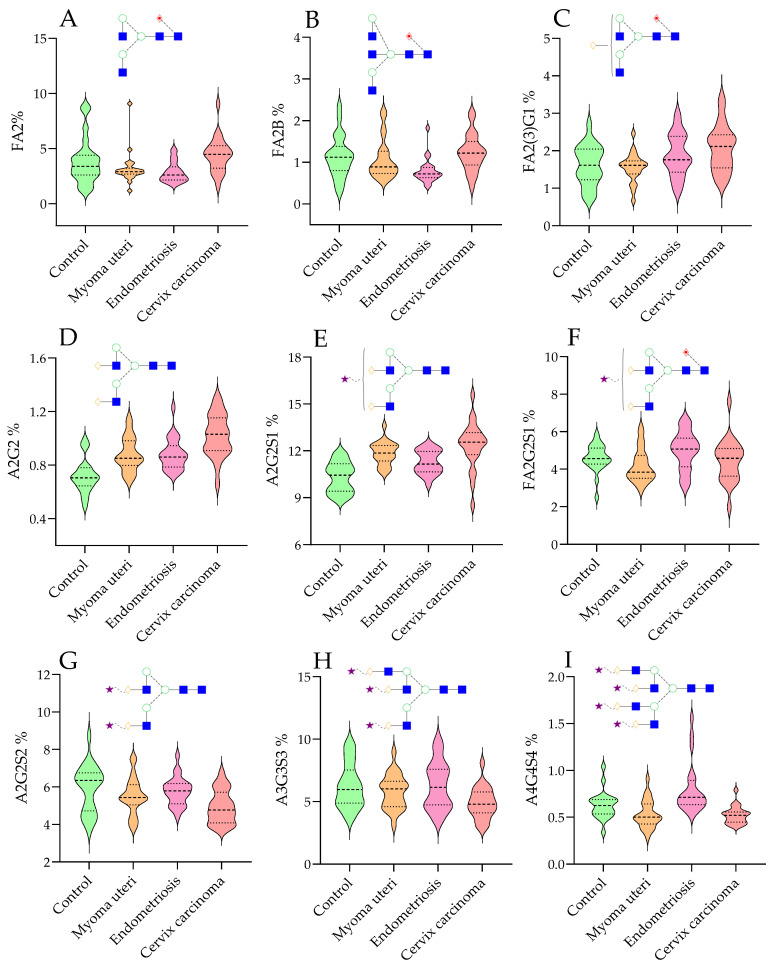
Significantly different serum N-glycan ratios in healthy control, myoma uteri, endometriosis, and cervical carcinoma patients by UPLC-HILIC-FLR ((**A**): FA2 fucosylated bi-antennary, (**B**): FA2BG1 fucosylated, mono-galactosylated and bisected bi-antennary, (**C**): FA2G1 fucosylated and mono-galactosylated bi-antennary, (**D**): FA2G2 fucosylated and bi-galactosylated bi-antennary, (**E**): A2G2S1 mono-sialylated and bi-galactosylated bi-antennary, (**F**): FA2G2S1 fucosylated mono-sialylated and bi-galactosylated bi-antennary, (**G**): A2G2S2 bi-sialylated and bi-galactosylated bi-antennary, (**H**): A3G3S3 tri-sialylated and tri-galactosylated tri-antennary, (**I**): A4G4S4 tetra-sialylated and tetra-galactosylated tetra-antennary).

**Table 1 ijms-26-04105-t001:** Baseline characteristics of the analyzed patient samples using Kruskall–Wallis test.

	Control	Sdev	Myoma Uteri	Sdev	Endometriosis	Sdev	Cervical Carcinoma	Sdev	*p* Value
Age	45.36	13.22	45.64	8.25	40.14	4.28	45.82	13.61	0.118
Urea	4.17	1.36	4.18	2.20	3.73	1.93	3.93	1.76	0.60
Creatinine	62.86	12.68	64.55	12.53	68.24	16.50	68.68	15.62	0.27
Estimated Glomerular Filtration Rate	85.00	9.31	88.50	13.75	85.38	15.52	85.64	15.65	0.47
Na	141.00	2.69	140.15	20.92	139.24	22.16	140.82	22.60	0.02
K	4.32	0.46	4.05	0.68	4.43	0.74	4.25	0.75	0.01
Glutamate Oxaloacetate Transaminase	23.41	13.57	20.70	10.18	17.38	9.09	32.32	26.75	0.07
Glutamate Pyruvate Transaminase	25.18	25.04	19.85	18.73	14.00	16.87	25.41	20.29	0.03
Gamma-Glutamyl Transferase	32.68	30.42	29.45	29.90	20.33	26.70	34.41	27.13	0.05
Alkaline Phosphatase	191.23	68.21	162.10	61.18	99.05	66.89	205.86	85.57	0.00
Lactate Dehydrogenase	365.00	90.08	343.65	90.27	331.54	91.19	357.68	98.51	0.65
C-Reactive Protein	7.33	6.36	3.38	5.72	21.47	26.21	3.99	23.10	0.01
White Blood Cells	7.92	1.79	7.01	1.90	7.72	2.15	7.33	2.30	0.41
Hemoglobin	134.05	11.19	117.75	24.78	124.43	24.43	130.23	24.67	0.02
Thrombocytes	255.18	110.11	293.00	90.23	298.19	87.45	271.73	84.43	0.05
International Normalized Ratio	1.08	0.31	0.98	0.25	0.96	0.22	0.96	0.21	0.01
Activated Partial Thromboplastin Time	30.25	4.93	29.66	5.75	30.24	5.78	29.75	5.70	0.89
Thrombin Time	18.33	2.56	17.73	3.00	17.63	3.08	17.51	3.06	0.68

**Table 2 ijms-26-04105-t002:** Kruskall–Wallis test of serum N-glycan ratios in healthy control, myoma uteri, endometriosis, and cervical carcinoma patients by UPLC-HILIC-FLR.

	Control	Sdev	Myoma Uteri	Sdev	Endometriosis	Sdev	Cervical Carcinoma	Sdev	*p* Value
FA2	3.88	2.07	3.20	1.49	2.85	0.92	4.42	1.59	0.00
M5	0.83	0.24	0.70	0.23	0.72	0.21	0.59	0.12	0.00
FA2B	1.14	0.51	1.07	0.48	0.79	0.30	1.23	0.45	0.00
FA2(6)G1	3.53	0.95	3.26	0.74	3.47	0.75	4.02	1.20	0.09
FA2(3)G1	1.61	0.53	1.55	0.38	1.88	0.57	2.08	0.62	0.01
FA2BG1	1.34	0.36	1.34	0.32	1.19	0.33	1.44	0.42	0.10
M6	0.91	0.30	0.99	0.25	1.08	0.24	1.01	0.17	0.19
A2G2	0.72	0.13	0.88	0.13	0.88	0.12	1.02	0.16	0.00
FA2G2	2.98	0.83	2.96	0.90	3.67	1.13	3.30	1.13	0.13
FA2BG2	1.02	0.18	1.07	0.17	1.06	0.18	1.06	0.21	0.81
FA2G1S1	0.80	0.21	0.70	0.10	0.79	0.20	0.77	0.16	0.15
A2G2S1	10.30	0.96	11.84	0.74	11.27	0.76	12.34	1.51	0.00
FA2G2S1	4.59	0.73	4.18	0.88	5.00	1.01	4.46	1.15	0.03
A2G2S2 (1)	5.93	1.26	5.52	0.99	5.77	0.76	4.94	0.89	0.01
A2G2S2 (2)	33.80	3.05	35.88	2.95	33.21	3.14	33.13	3.98	0.03
FA2G2S2	5.35	1.42	5.12	0.64	5.08	0.81	5.20	1.05	0.99
FA2BG2S2	2.29	1.08	1.82	0.74	1.82	0.56	1.94	0.82	0.23
A2BG3S2	1.69	0.43	1.77	0.39	1.88	0.54	1.74	0.41	0.75
A3G3S2	1.46	0.36	1.64	0.35	1.53	0.35	1.62	0.33	0.32
FA3G3S2	1.19	0.32	1.09	0.27	1.25	0.20	1.22	0.31	0.25
A3G3S3 (1)	0.40	0.09	0.39	0.04	0.40	0.10	0.40	0.09	0.98
A3G3S3 (2)	6.25	1.78	5.80	1.38	6.29	1.90	4.90	1.25	0.03
A3G3S3 (3)	0.34	0.13	0.27	0.07	0.33	0.10	0.26	0.09	0.01
FA3G3S3 (1)	0.60	0.29	0.55	0.13	0.60	0.29	0.51	0.19	0.49
A3G3S3 (4)	1.79	0.60	1.90	0.63	1.73	0.49	1.69	0.45	0.54
FA3G3S3 (2)	2.89	1.08	2.57	1.30	2.73	1.31	2.81	1.46	0.88
FA3G3S3 (3)	0.90	0.17	0.78	0.12	0.96	0.24	0.76	0.15	0.00
A4G4S3	0.35	0.07	0.29	0.10	0.36	0.12	0.30	0.11	0.05
A4G4S4 (1)	0.50	0.21	0.33	0.11	0.60	0.23	0.33	0.15	0.00
A4G4S4 (2)	0.63	0.14	0.53	0.15	0.80	0.27	0.52	0.09	0.00

## Data Availability

The generated data can be requested from the corresponding author.

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
