# Peer review of "Comparative Analysis of Serum N-Glycosylation in Endometriosis and Gynecologic Cancers"

_ijms, 2025, doi:10.3390/ijms26094105_

Round 1

Reviewer 1 Report

Comments and Suggestions for Authors

Peer Review Report

Manuscript Title: Comparative Analysis of Serum N-glycosylation in Endometriosis and Gynecologic Cancers

Authors: Róbert Pásztor, Béla Viskolcz, Csaba Oláh, and Csaba Váradi

This study explores serum N-glycosylation profiles in endometriosis, myoma uteri, and cervical cancer using HILIC-UPLC and mass spectrometry. The topic is highly relevant, and the identification of disease-specific glycan signatures could be valuable for biomarker discovery. The methodology is well-executed, but some critical areas require improvement, including statistical rigor, interpretation of biological mechanisms, and clarity of data presentation.

Abstract Review

  1. The abstract lacks a clear problem statement or hypothesis. It should explicitly state whether the goal is biomarker discovery, mechanistic insights, or disease differentiation.
  2. Lack of specific results in glycosylation changes. The abstract mentions glycosylation shifts but does not specify which glycan patterns were significantly altered.
  3. Better clarity on implications and limitations. The clinical and diagnostic implications are implied but not explicitly stated.

Major Concerns

  1. Lack of a Clear Hypothesis and Study Rationale
  • The manuscript lacks a clearly stated hypothesis or research question. It appears exploratory rather than hypothesis-driven, which weakens the scientific impact (specifically in lines 57-61).
  1. Sample collection
  • The study does not provide sufficient details on how and when the samples were collected. Inclusion and exclusion criteria are missing, making it unclear how participants were selected. Recommendation: Clearly outline the sample collection process, including: Inclusion/exclusion criteria; Collection protocols (e.g., fasting state, storage conditions); Potential confounders (e.g., hormonal treatment, comorbidities).
  1. Results and Discussion
  • When presented together without subtitles, it becomes difficult to follow.
  1. Limited Discussion of Biological Mechanisms
  • The study effectively identifies glycosylation shifts but does not sufficiently explain why these changes occur in different disease states. Example: Why does A2G2 increase in cervical carcinoma? What is the biological significance of decreased bisected glycans in endometriosis?
  1. Data Presentation
  • Table 1. Introduce the parameters clearly; do not use only abbreviations. Also, highlight significant p-values and specify the tests used.
  • Table S1. The significant results must be included in the main paper, not only in the S1 material.
  • Figures 2 and 3:
    • If these results are significant, where is the p-value? Additionally, the authors mentioned that 14 N-glycan structures showed significantly different relative abundances among the four groups according to the Kruskal–Wallis test. However, the figure only represents 9 structures. I would prefer to see all 14 included here.

Minor Concerns

  1. Terminology and Clarity
  • Abbreviations must be fully explained the first time they are used (lines 58-59).
  • The study uses specialized glycomic terminology (e.g., FA2, A2G2S1) without sufficient explanation for non-experts. Recommendation: Provide a supplementary table with full glycan names, functions, and biological relevance.
  • What do you mean by “healthy controls”?
  1. Interpretation of ROC Curves
  • ROC analysis shows that combinations of glycans improve classification, but no multivariate model is proposed. Recommendation: Use logistic regression or machine learning to test whether a glycan panel improves diagnostic accuracy over single markers.

This study provides novel insights into glycosylation differences in gynecologic diseases, but improvements are needed in biological interpretation, statistical rigor, and data presentation. Addressing these concerns will significantly enhance the manuscript's impact and clarity.

Would be happy to review a revised version incorporating these suggestions.

Author Response

Abstract Review

  1. The abstract lacks a clear problem statement or hypothesis. It should explicitly state whether the goal is biomarker discovery, mechanistic insights, or disease differentiation.

Response 1: Thank you for your suggestion. It has been added to the abstract.

  1. Lack of specific results in glycosylation changes. The abstract mentions glycosylation shifts but does not specify which glycan patterns were significantly altered.

Response 2: Thank you for your comment. The abstract has been modified accordingly.

  1. Better clarity on implications and limitations. The clinical and diagnostic implications are implied but not explicitly stated.

 Response 3: It has been added that this study serves as a proof of concept.

Major Concerns

  1. Lack of a Clear Hypothesis and Study Rationale
  • The manuscript lacks a clearly stated hypothesis or research question. It appears exploratory rather than hypothesis-driven, which weakens the scientific impact (specifically in lines 57-61).

 Response 4: We aimed to examine the biomarker potential of protein glycosylation which has been added to the main text. Thank you for your suggestion.

  1. Sample collection
  • The study does not provide sufficient details on how and when thesamples were collected. Inclusion and exclusion criteria are missing, making it unclear how participants were selected. Recommendation: Clearly outline the sample collection process, including: Inclusion/exclusion criteria; Collection protocols (e.g., fasting state, storage conditions); Potential confounders (e.g., hormonal treatment, comorbidities).

 Response 5:

Thank you for your comment. We have modified the corresponding paragraph.

All disease groups were confirmed through histological analysis, as each patient underwent surgical treatment. Patients were selected based on the absence of comorbidities, and healthy controls reported no gynecological complaints.The preparation of serum samples is a critical step in various laboratory analyses, particularly in clinical and research settings. Blood samples were collected using sterile vacutainer tubes without anticoagulants. After collection, the samples were allowed to clot at room temperature for approximately 30 minutes, facilitating the separation of serum from cellular components. Subsequently, samples were centrifuged at 2000 RPM for 10-15 minutes. The serum was carefully pipetted into new sterile tubes, ensuring that the clot remained undisturbed. For storage, samples were frozen at -80°C, depending on the requirements of the intended analysis.

  1. Results and Discussion
  • When presented together without subtitles, it becomes difficult to follow.

Response 6: Thank you for your suggestion. Subtitles have been added to the results and discussion.

  1. Limited Discussion of Biological Mechanisms
  • The study effectively identifies glycosylation shiftsbut does not sufficiently explain why these changes occur in different disease states. Example: Why does A2G2 increase in cervical carcinoma? What is the biological significance of decreased bisected glycans in endometriosis?

Response 7: As it has been mentioned at the limitations of this study, the low sample size does not allow for far-reaching conclusions and we have tried to be careful with this.

The increase in A2G2 levels in cervical cancer can be attributed to several factors. Cancer cells often exhibit abnormal glycosylation, which can result in the overexpression of specific glycan structures like A2G2. This alteration is linked to changes in glycosyltransferase enzyme expression, which modifies how sugars are attached to proteins and lipids on the cell surface. The presence of A2G2 has been associated with enhanced tumor aggressiveness and metastatic potential. It may play a role in promoting cell adhesion, migration, and invasion, which are critical processes in cancer progression. A2G2 may help tumor cells evade the immune system by masking tumor antigens or modulating immune cell interactions, thereby promoting tumor survival.

Regarding the biological significance of decreased bisected glycans in endometriosis:

Inflammation and Immune Response: Bisected glycans are typically involved in modulating immune responses. A decrease in these glycans may lead to altered immune signaling, contributing to the chronic inflammation observed in endometriosis. This may affect the local immune environment and influence disease progression. Cellular Adhesion and Migration: Bisected glycans play a role in cell-cell and cell-matrix interactions. Their decrease may impair the normal adhesion and migration of endometrial cells, which can contribute to the ectopic implantation of endometrial tissue characteristic of endometriosis.

  1. Data Presentation
  • Table 1. Introduce the parameters clearly; do not use only abbreviations. Also, highlight significant p-values and specify the tests used.

Response 8: Thank you for your suggestion. It has been modified according to the suggestion.

  • Table S1.The significant results must be included in the main paper, not only in the S1 material.

Response 9: The Table S1 has been moved to the manuscript.

  • Figures 2 and 3:
    • If these results are significant, where is the p-value? Additionally, the authors mentioned that 14 N-glycan structures showed significantly different relative abundances among the four groups according to the Kruskal–Wallis test. However, the figure only represents 9 structures. I would prefer to see all 14 included here.
    • Response 10:

p values are listed now in Table 2.

We acknowledge that our current figure includes only 9 of the 14 N-glycan structures identified as significantly different. The decision to limit the number of structures displayed was made with the intent to maintain clarity and visual impact. Including all 14 structures in a single figure would either result in a very crowded layout or require a reduction in size that could compromise visibility and interpretation.

Minor Concerns

  1. Terminology and Clarity
  • Abbreviations must be fully explained the first time they are used (lines 58-59).

Response 11:

  • Thank you for your comment regarding the necessity of fully explaining abbreviations upon their first use. We acknowledge the importance of clarity and accessibility in our writing.We would like to point out that the abbreviations mentioned in lines 58-59 are indeed fully explained in the abstract of our manuscript. We believe that this placement allows readers to understand the terms before encountering them in the main text.However, to enhance readability and ensure that all readers can easily follow our work without having to refer back to the abstract, we will revise the manuscript to include full explanations of the abbreviations at their first occurrence in the main body of the text. This will ensure that all relevant information is immediately accessible and improve the overall clarity of our presentation.Thank you for highlighting this point, and we appreciate your suggestions for improving the manuscript.
  • The study uses specialized glycomic terminology (e.g., FA2, A2G2S1) without sufficient explanation for non-experts. Recommendation: Provide a supplementary table with full glycan names, functions, and biological relevance.

Response 12:

We have generated an explanatory Figure in the Supplementary Material.

  • What do you mean by “healthy controls”?

Response 13: Healthy Controls refer to a group of individuals who are had no gyneologycal complaint. Typically visitin gynecologist as part of occupational health examination.

  1. Interpretation of ROC Curves
  • ROC analysis shows that combinations of glycans improve classification, but no multivariate model is proposed. Recommendation:Use logistic regression or machine learning to test whether a glycan panel improves diagnostic accuracy over single markers.

 Response 14:

Thank you for your insightful feedback regarding the ROC analysis and the potential for improved classification using glycan combinations. We appreciate your suggestion to employ logistic regression or machine learning to evaluate whether a glycan panel enhances diagnostic accuracy compared to individual markers.We acknowledge the limitation posed by the low number of samples in our current study, which may impact the robustness of our findings.

To address this, we plan to take the following steps:

Expand Sample Size: We recognize that a larger sample size is essential for more reliable model training and validation. We will explore opportunities to collaborate with other institutions or utilize existing datasets to increase our sample pool.

Utilize Cross-Validation: In our analysis, we will implement cross-validation techniques to maximize the use of our available data. This approach will help ensure that our models are generalizable and mitigate the risk of overfitting, even with a limited number of samples.

Logistic Regression and Machine Learning Models: We will develop both logistic regression models and machine learning algorithms to compare the diagnostic performance of glycan panels against single markers. This dual approach will allow us to assess not only the accuracy but also the potential improvements in sensitivity and specificity offered by the glycan combinations.

Feature Selection: We will employ feature selection techniques to identify the most informative glycans for inclusion in our models. This will help streamline our analysis and focus on the most relevant markers.

Follow-Up Studies: If initial results are promising, we will consider conducting follow-up studies with a larger cohort to further validate our findings and explore the clinical implications of using glycan panels in diagnostic settings.

We appreciate your recommendations and believe that addressing the sample size limitation while employing robust statistical methods will enhance the credibility of our findings and contribute to the growing understanding of glycan-based diagnostics.

Reviewer 2 Report

Comments and Suggestions for Authors

This article focuses on comparative analysis of serum N-glycosylation in Endometriosis and Gynecologic cancer to identify N-glycan biomarkers that can differentiate between healthy and various gynecologic pathologies. The authors analyzed the serum N-glycome of the 22 healthy controls, 22 myoma uteri patients, 22 endometriosis patients, and 22 cervical carcinoma patients using the HILIC-HPLC with fluorescence and MS detection. The authors identified characteristic glycosylation-based signatures in each disease group when compared to control groups. They also reported consistently elevated levels of specific bi-antennary glycans (A2G2 and A2G2S1). This study provides a promising strategy for identifying non-invasive indicators for gynecologic conditions. The authors have done a good job of identifying the possible biomarkers and advancing the current knowledge in the field of glycobiology. However, I do have some questions and comments. I would recommend this work for publication after minor revisions.

Comments:

Table 1: The authors talk about the elevated levels of CRP in endometriosis when compared to the control and other diseased states. However, the level of CRP is also decreased in Myoma uteri and Cervix carcinoma as compared to control group. I was wondering if there is a reasoning for that. Similarly for the levels of ALP are also significantly low in endometriosis. Could the authors provide the reasoning behind these as well.

Line 108: The authors introduce the abbreviations like G0F. Can the authors define all the abbreviations?

Line 112/113: The authors talk about the levels of FA2G2. I would suggest authors to use the figure or table reference every time they talk about the data. Sometimes, it’s hard to follow with these many abbreviations.

Figure 2: The reference glycan structures on the figures are very small and is hardly visible. Can the authors increase the size so that it is clearly visible?

Line 194-197: Include the figure reference, so that it is easy to follow.

The authors have mentioned about some of the limitations such as sample size, lack of independent validation. However, have the authors thought about the influence of O-linked glycosylation. O-linked glycosylation is also altered in many diseased states. I was wondering if the authors have thought about studying the changes or differences in O-glycans and if that can also be studied as potential biomarkers.

Author Response

Comments:

Comment 1:

Table 1: The authors talk about the elevated levels of CRP in endometriosis when compared to the control and other diseased states. However, the level of CRP is also decreased in Myoma uteri and Cervix carcinoma as compared to control group. I was wondering if there is a reasoning for that. Similarly for the levels of ALP are also significantly low in endometriosis. Could the authors provide the reasoning behind these as well.

Response 1:

The differences in CRP and ALP levels among endometriosis, myoma uteri, and cervix carcinoma can be understood through the lens of inflammation, immune response, and tissue-specific factors. Elevated CRP levels in endometriosis reflect its inflammatory nature, while decreased levels in myoma uteri and cervix carcinoma suggest a different pathophysiological context. Similarly, the reduced ALP levels in endometriosis may reflect underlying metabolic or physiological changes associated with the disease. Further research is needed to fully elucidate these relationships and their clinical implications.

Comment 2:

Line 108: The authors introduce the abbreviations like G0F. Can the authors define all the abbreviations?

Response 2:

Thank you for your comment. As it is described G0F is analogous to FA2 which is a fucosylated bi-antennary with no galactose. Please note that some laboratories are using different nomenclatures thaat is the reason why we have clarified that FA2=G0F.

Comment 3:

Line 112/113: The authors talk about the levels of FA2G2. I would suggest authors to use the figure or table reference every time they talk about the data. Sometimes, it’s hard to follow with these many abbreviations.

Response 3:

Thank you for your comment. We have modified the text according to the suggestion.

Comment 4:

Figure 2: The reference glycan structures on the figures are very small and is hardly visible. Can the authors increase the size so that it is clearly visible?

Response 4:

It seems to be a converting issue, I guess the reviewer downloaded the pdf version which contains really small structures although it seems fine in the word version. We are going to make sure that it has the same size after converting.

Comment 5:

Line 194-197: Include the figure reference, so that it is easy to follow.

Response 5:

Thank you for the suggestion. It has been corrected.

Comment 6:

The authors have mentioned about some of the limitations such as sample size, lack of independent validation. However, have the authors thought about the influence of O-linked glycosylation. O-linked glycosylation is also altered in many diseased states. I was wondering if the authors have thought about studying the changes or differences in O-glycans and if that can also be studied as potential biomarkers.

Response 6:

In our laboratory we are traditionally analyzing N-glycosylation. We are actually interested in the analysis of O-glycosylation although the analysis and sample preparation for O-glycosylation and N-glycosylation differ significantly due to their unique characteristics and structural properties. For the analysis of O-glycans additional enzymes (like sialidases or O-glycanases) may be necessary to release O-glycans from the protein backbone.

Round 2

Reviewer 1 Report

Comments and Suggestions for Authors

The authors have carefully revised the manuscript following the comments provided. The study addresses a highly relevant topic, and identifying disease-specific glycan signatures offers valuable potential for biomarker discovery. The methodology is solid, and the revisions have improved the manuscript substantially. Therefore, I recommend the paper for acceptance.